# Transcriptome Sequencing and Metabolite Analysis Revealed the Single and Combined Effects of Microplastics and Di-(2-ethylhexyl) Phthalate on Mouse Liver

**DOI:** 10.3390/ijms26104943

**Published:** 2025-05-21

**Authors:** Jiabin Zhang, Yangcheng Li, Yihan Wang, Zeyu Li, Xiaolei Li, Hongxia Bao, Jiakui Li, Donghai Zhou

**Affiliations:** College of Animal Medicine, Huazhong Agricultural University, Wuhan 430070, China; jiabinzhanghzau@163.com (J.Z.); ylee6576@gmail.com (Y.L.); wangyh2525@163.com (Y.W.); ljkzhx@163.com (Z.L.); lixiaolei0516@163.com (X.L.); 15527569823@163.com (H.B.)

**Keywords:** microplastics, di-(2-ethylhexyl) phthalate, combined toxicity, transcriptomic, metabolomic, mice

## Abstract

The widespread use of plastics has led to a substantial increase in plastic waste, resulting in the dissemination of plastic debris throughout ecosystems and posing significant threats to biota. Bis(2-ethylhexyl) phthalate (DEHP), a commonly used plasticizer, enhances plastic flexibility but may also exert subtle toxic effects. This study aimed to investigate the potential toxicological impacts and underlying mechanisms of microplastics (MPs), di-(2-ethylhexyl) phthalate (DEHP), and their combined exposure (MPs + DEHP) on oxidative stress, apoptotic damage, transcriptomic alterations, and metabolic disturbances in mice. The results demonstrated that exposure to MPs, DEHP, and MPs + DEHP impaired the antioxidant defense system and reduced overall antioxidant capacity. Concurrently, all three exposure conditions significantly increased biochemical markers, particularly those associated with liver dysfunction, prompting further analysis of hepatic tissues. Histopathological examination revealed apoptotic damage in hepatocytes. Integrated transcriptomic and metabolomic analyses indicated that exposure to MPs, DEHP, and MPs + DEHP disrupted carbohydrate, amino acid, and lipid metabolism, induced the expression of genes related to hepatocarcinogenesis, and impaired purine metabolism. Moreover, MP and DEHP exposure aggravated hepatic apoptosis and inflammatory responses via activation of the PI3K/AKT signaling pathway, thereby eliciting notable biotoxic effects. These findings provide new scientific evidence regarding the individual and combined toxicological effects of MPs and the plastic additive DEHP on living organisms.

## 1. Introduction

With the rapid development of socio-economics and continuous advancements in science and technology, the widespread application and significant contributions of plastic products in modern society have become increasingly prominent [1]. However, in recent years, plastic pollution has become increasingly severe, posing serious threats to various ecological environments such as terrestrial, wetland, marine, and atmospheric systems, making it a global challenge that urgently needs to be addressed in the field of environmental protection [2,3]. Microplastics (MPs), defined as plastic fragments with a diameter of less than 5 mm, primarily originate from the degradation of widely used plastic products [4,5,6]. Under long-term physicochemical factors such as photo-oxidation, plastic materials undergo gradual degradation, eventually transforming into microplastics [7] and may further fragment into even smaller particles [8]. These microplastic can accumulate in aquatic ecosystems and terrestrial soil environments through bioaccumulation. Notably, MPs have been widely detected in air, water, and soil environments [9,10]. Due to their small size, MPs can be ingested by organisms and accumulate in tissues, causing both physical mechanical injury and chemical toxicity [11].

Extensive studies have shown that accumulation of MPs in the digestive systems of experimental animals, such as zebrafish and mice, can induce a range of pathological alterations, including mucosal damage, inflammatory responses, and metabolic disturbances [12,13]. For instance, polystyrene microplastics (PS-MPs) have been shown to cause oxidative stress in mouse skeletal muscle tissues [14], promote programmed muscle cell death [15], and induce oxidative damage and mitochondrial dysfunction in osteoblasts [16]. At the nanoscale, particles may even lead to Parkinson’s disease (PD)-like neurodegeneration in mice [17]. Research by Lu et al. demonstrated that PS-MP exposure significantly inhibits weight gain in mice and results in pathological alterations in liver and intestinal tissues [18]. Additionally, MPs may further aggravate adverse health effects by disrupting the gut microbiota [19]. Biomonitoring studies have detected microplastics in human blood, placenta, urine, and feces [20,21,22,23]. More alarmingly, microplastics can bioaccumulate through trophic transfer in the food web and ultimately enter the human body. Previous studies have identified the presence of microplastics in human testes and semen [24], and more recent research has also detected microplastics in follicular fluid [25], posing a potential threat to human reproductive health.

Importantly, MPs possess the capacity to adsorb or release other pollutants, including organic contaminants, heavy metals, and antibiotics, which may exert synergistic toxic effects on organisms [26,27]. Among them, phthalate esters (PAEs), as the most commonly used plastic additives, represent a class of organic pollutants with significant environmental hazards [28]. Di(2-ethylhexyl) phthalate (DEHP), the main component of plasticizers, is widely employed to enhance the elasticity and flexibility of plastic products. Present in a free state within plastics, DEHP primarily associates with the polymer matrix via hydrogen bonding and van der Waals forces, making it highly prone to leaching and volatilization into the surrounding environment [29]. Once released, DEHP may enter the food chain and bioaccumulate, becoming a ubiquitous global pollutant [30,31] and thereby posing considerable risks to atmospheric, terrestrial, and marine environmental safety [32]. Studies have revealed that DEHP and its metabolites exhibit endocrine-disrupting properties and can impair reproductive health, hepatic metabolism, immune function, and nervous system [33]. DEHP has been widely detected in water, meat, eggs, and dairy products, raising increasing public health concerns [34]. Furthermore, in many countries, the DEHP content in polyvinyl chloride (PVC) can reach up to 60%, and with time, DEHP leaches from polymers into the environment. For instance, concentrations in landfill leachate may reach as high as 394.4 μg/L [35]. Both MPs and DEHP have been shown to induce hepatotoxicity and nephrotoxicity [36,37], as well as reproductive toxicity [38,39], and can disrupt endocrine functions [40]. DEHP has also been found to suppress spontaneous tail movements in zebrafish and decrease overall activity levels [41]. In mice, adolescent exposure to DEHP may result in increased anxiety and long-term behavioral alterations [42]. Although DEHP is recognized as a reproductive toxin and endocrine disruptor [43], studies focusing on the combined toxicological mechanism of MPs and DEHP remain limited.

While several studies have elucidated the individual toxic effects of MPs and DEHP in murine models, data on their combined effects remain scarce. Investigating the co-exposure effects of MPs and DEHP will enhance our understanding of their potential synergistic toxicity in biological systems. Humans are exposed to MPs and DEHP through various routes, including ingestion, inhalation, and dermal absorption, with ingestion considered the primary pathway in the general population [44]. In the European Union (EU), urinary biomonitoring of DEHP and its metabolites estimated human exposure to DEHP at 17 μg/kg body weight/day. Exposure assessments based on the EUSES (European Union System for the Evaluation of Substances) model estimated adult exposure between 2 and 67 μg/kg body weight/day, while children’s exposure levels were substantially higher, ranging from 20 to 312 μg/kg body weight/day [45]. Human exposure to primary plastic particles smaller than 3 μm or 100 μm has been estimated at cumulative levels of up to 19,000 mg/year/L [46,47]. It is estimated that less than 10% of plastic waste consists of polystyrene particles, with average individual intake of PS particles ranging from 0 to 19 μg/mL [48]. In this study, the experimental period in mice was designed to correspond with the human childhood developmental stage [49]. Based on human exposure data and murine dietary intake, mice were provided with food containing environmentally relevant concentrations of DEHP (200 mg/kg feed) and MPs in drinking water (10 mg/L). The objectives of this study are as follows: (1) Elucidate the toxic effects of individual and combined exposure to MPs and DEHP on oxidative stress responses and tissue apoptosis in mice. (2) Systematically assess the impacts of MPs and DEHP—individually and in combination—on metabolic homeostasis and gene regulation using integrated metabolomics and transcriptomics approaches. (3) Clarify the molecular mechanisms through which individual and co-exposure to MPs and DEHP induce hepatocellular injury and inflammatory responses via modulation of the hepatic PI3K/AKT apoptotic signaling pathway.

## 2. Results

### 2.1. Effects of MPs, DEHP, and MPs + DEHP on Antioxidant Biomarkers in Mice

As shown in Figure 1, serum biochemical indicators such as AST, ALT, UREA, CREA, CK, and ALP were significantly increased in the MP, DEHP and MP + DEHP groups compared with the normal control group (Figure 1A). In terms of antioxidant activity, LDH and MDA contents were significantly increased in the MP, DEHP and MP + DEHP groups compared with the Con group. However, MPs, DEHP and MPs + DEHP inhibited the ability of T-AOC and the activities of CAT and SOD (Figure 1B).

### 2.2. Effects of MPs, DEHP, and MPs + DEHP on Liver Tissue in Mice

After the mice were dissected, we observed the liver tissue and found that the liver tissue of the Con and MP groups was red, while the liver tissue of the DEHP and MP + DEHP groups was obviously deepened and black–red (Figure 2A). To assess the effects of MPs and DEHP on mouse liver tissue, histological changes in the livers of MP- and DEHP-exposed mice were observed using H&E staining (Figure 2B). Both the MP and DEHP monotherapy groups showed hyperemic lesions (indicated by the black arrow in the Figure 2B) and a trend toward cytoplasmic vacuolization (indicated by the red arrow in the Figure 2B) compared with the Con group. The MP + DEHP group showed obvious hyperemia (indicated by the black arrow in the Figure 2B), cell necrosis and obvious vacuolization (indicated by the red arrow in the Figure 2B), and a tendency to connective tissue hyperplasia.

### 2.3. Effects of MPs, DEHP, and MPs + DEHP on Transcriptomic Profiles in Mice

To further investigate the potential responses of mice to MPs, DEHP, and MPs + DEHP, we performed transcriptome analysis of mouse liver tissues collected at the endpoint. PCA analysis showed that three biological replicates clustered within each group, with significant differences between groups (Figure 3A). The Q30 and GC contents were >95.05% and 48.83%, respectively (Appendix A), and about 87.51–89.83% of the total mapped reads were uniquely mapped (Appendix A). The above results indicate that high-quality sequence data were obtained for further analysis. Based on the gene expression in different groups, Venn analysis was performed to identify co-expressed and specifically expressed genes between the groups. In the three comparisons (MPs/Con, DEHP/Con, MPs + DEHP/Con) among the groups, 176, 2330, and 1844 differentially expressed genes were identified, respectively (Figure 3B). DEHP/Con resulted in a substantial number of differentially expressed genes (1218 up-regulated and 1112 down-regulated genes). A relatively small number of differentially expressed genes were significantly altered by MPs (74 up-regulated and 102 down-regulated). Meanwhile, MPs + DEHP/Con induced differentially expressed genes (1039 up-regulated and 805 down-regulated differentially expressed genes) (Figure 3D and Appendix A). KEGG enrichment analysis showed that the differentially expressed genes were enriched in specific functions or pathways. Specifically, 38 pathways were significantly induced in the MP group (*p* < 0.05), for example, the cancer pathway (mmu05200, 12 DEGs), HIF-1 signaling pathway (mmu04066, 5 DEGs), chemical carcinogenesis-reactive oxygen species (mmu05208, 5 DEGs), glycolysis/gluconeogenesis (mmu00010, 5 DEGs), glutathione metabolism (mmu00480, 5 DEGs), and others (Figure 3C). Similarly, 67 pathways were significantly affected in the DEHP group (*p* < 0.05). For instance, hepatocellular carcinoma (mmu05225, 31 DEGs), cancer pathways (mmu05200, 72 DEGs), peroxisome (mmu04146, 37 DEGs), Alzheimer’s disease (mmu05010, 53 DEGs), and non-alcoholic fatty liver disease (mmu04932, 27 DEGs) were among those affected (Figure 3C). Of the 65 significantly altered pathways (*p* < 0.05) in the MP + DEHP group, the most prominent included cancer pathways (mmu05200, 62 DEGs), alcoholic liver disease (mmu04936, 21 DEGs), fluid shear stress and atherosclerosis (mmu05418, 23 DEGs), the PI3K-Akt signaling pathway (mmu04151, 39 DEGs), and the MAPK signaling pathway (mmu04010, 36 DEGs) (Figure 3C).

### 2.4. Effects of MPs, DEHP, and MPs + DEHP on Metabolomics Profiles in Mice

To further investigate the toxicological effects of MP, DEHP, and MP + DEHP exposure on the metabolic response of liver tissue in mice, UHPLC-MS/MS was used for metabolic profiling. Mouse metabolomics QC samples were evaluated in both positive and negative ion modes, and for the overall data, an RSD < 0.3 and a cumulative proportion of peaks > 70% were considered acceptable (Figure 4A). Using PLS-DA, we observed significant separation between the Con group and the MP, DEHP, and MP + DEHP groups (Figure 4B), indicating that exposure to MPs, DEHP, and MPs + DEHP significantly altered liver tissue metabolism in mice. A total of 1082 metabolites (433 positive ions and 649 negative ions) were identified. Specifically, there were 134 (77 up-regulated, 57 down-regulated), 215 (118 up-regulated, 97 down-regulated), and 204 (106 up-regulated, 98 down-regulated) in the MP, DEHP, and MP + DEHP groups compared to the control group (Figure 4C). Taken together, these findings indicate that exposure to MPs, DEHP, and MPS-DEHP caused significant changes in liver tissue metabolism in mice. KEGG pathway enrichment analysis of DEMs indicated that DEMs were involved in different metabolic pathways in response to different exposure environments of MPs, DEHP, and MPs + DEHP. The results showed that DEMs in the MP group were involved in cofactor and vitamin metabolism, amino acid metabolism, lipid metabolism, and cancer overview (Figure 4D); DEMs in the DEHP and MPs + DEHP groups were involved in lipid metabolism, carbohydrate metabolism, amino acid metabolism, and the digestive system (Figure 4E,F).

### 2.5. Integrated Analysis of Transcriptomic and Metabolic Profiles in Mouse Liver Tissue

Further screening of differentially expressed genes (DEGs) and differentially expressed metabolites (DEMs) was conducted to identify significantly enriched pathways in the comparison groups MPs/Con, DEHP/Con, and MPs + DEHP/Con, as well as in the corresponding metabolite datasets. In addition, a more detailed investigation into the correlation between DEGs and DEMs was performed. As illustrated in Figure 5A–C, the union of tertiary pathways, significantly enriched in both gene and metabolite datasets for the three comparison groups, was analyzed, and the top 10 enriched pathways were selected for each group. These results revealed a relatively strong correlation between DEGs and DEMs in the DEHP/Con and MPs + DEHP/Con groups. Notably, exposure to DEHP and combined exposure to MPs + DEHP exerted a greater impact at the transcriptional level in mice, indicating that the toxicological mechanisms of MPs and DEHP may differ. Through integrated transcriptomic and metabolomic analysis, the relationships between DEGs and DEMs within common KEGG-enriched pathways were examined. The findings suggest that the metabolic responses to MP, DEHP, and MP + DEHP exposure are mediated through distinct pathways. Specifically, in the MP group, the most significantly affected pathways were associated with amino acid and lipid metabolism. In the DEHP group, the predominant pathways involved carbohydrate metabolism, lipid metabolism, the metabolism of cofactors and vitamins, the metabolism of other amino acids, and mineral absorption. For the MP + DEHP co-exposure group, the most enriched pathways included the PI3K-Akt signaling pathway, the metabolism of cofactors and vitamins, amino acid metabolism, and cofactor biosynthesis.

### 2.6. Effects of MPs, DEHP, and MPs + DEHP on the PI3K/AKT Pathway in Mouse Liver Cells

The toxic effects of MP, DEHP, and MP + DEHP co-exposure on the liver organs of mice were further investigated. We then examined the expression of proteins and genes related to apoptotic injury in mouse liver tissues (Figure 6). This study found that compared with the Con group, the protein levels and mRNA expression of PI3K, AKT, and Bcl-2 in the MP group were significantly down-regulated, and the protein levels and mRNA expression of TNF-α, Bax, and Caspase 3/8 were significantly up-regulated. However, the protein level and mRNA expression of IL-6 and Caspase 9 were not obvious. The protein levels and mRNA expressions of PI3K, AKT, and Bcl-2 in the DEHP group were significantly down-regulated, while the protein levels and mRNA expressions of IL-6, TNF-α, Bax, and Caspase 3/8/9 were significantly up-regulated. The protein and mRNA expression levels of PI3K, AKT, and Bcl-2 in the MP + DEHP group were significantly down-regulated, while the protein and mRNA expression levels of IL-6, TNF-α, Bax, and Caspase 3/8/9 were significantly up-regulated. Based on these results, we concluded that MP, DEHP, and MP + DEHP exposure could cause a certain inflammatory response and apoptotic damage to the liver tissue of mice, and the effect of MP + DEHP co-exposure was more significant.

## 3. Discussion

This study explored the toxicological effects of MPs, DEHP, and their combination in mice, focusing on oxidative stress, transcriptomic and metabolomic alterations, and liver apoptosis. Exposure to MPs, DEHP, and their combination led to significant disruptions in antioxidant biomarkers, gene expression profiles, and metabolic pathways, indicating the potential risks associated with chronic exposure to these pollutants. Notably, all three treatments impaired the antioxidant defense system, triggered inflammatory responses, and induced liver apoptosis, likely due to disturbances in amino acid and lipid metabolism. These findings provide novel insights into the toxicological mechanisms of MPs and plastic additives, underscoring the urgent need to address their potential threats to human health and ecological systems.

### 3.1. Effects of MP and DEHP Exposure on the Mouse Liver

In this study, chronic exposure to MPs, DEHP, and their combination resulted in marked alterations in serum biochemical indices and liver histopathology in mice. All exposure groups showed significantly elevated biochemical markers, and there were also significant differences between the DEHP and MP groups, as well as between the DEHP and combined exposure groups. Notably, elevated levels of AST, ALT, ALP, and urea were detected, which are classical biomarkers indicative of hepatocellular injury, cholestasis, and hepatic metabolic dysfunction, respectively [50,51,52]. Consistent with these biochemical changes, histological analysis revealed significant hepatic lesions, including vascular congestion, necrosis, and cytoplasmic vacuolization of hepatocytes. These findings demonstrate that all three exposure conditions induced liver damage, with the combined exposure potentially eliciting additive or synergistic toxic effects.

Moreover, integrated transcriptomic and metabolomic analyses revealed that DEGs and metabolites were significantly enriched in pathways related to bile secretion, further highlighting the disruption of hepatobiliary function. Collectively, these results indicate that MPs and DEHP, whether individually or in combination, compromise liver integrity and function through coordinated biochemical and structural perturbations.

### 3.2. Effects of MP and DEHP Exposure on Oxidative Stress and Inflammatory Responses in Mice

Oxidative stress is a well recognized toxicological mechanism triggered by environmental pollutants. When organisms are exposed to toxic substances in the environment that induce excessive oxidative stress, the risk of developing a wide range of diseases increases significantly [53]. In this study, exposure to MPs, DEHP, and their combination markedly impaired the antioxidant defense system in mice, as evidenced by decreased activities of key antioxidant enzymes—SOD, CAT, and T-AOC—accompanied by elevated levels of LDA and lipid peroxidation marker MDA [54,55]. These findings suggest an overproduction of ROS, leading to oxidative imbalance and cellular damage.

Previous studies have demonstrated that ROS can initiate lipid peroxidation, resulting in structural damage to cellular membranes and triggering apoptosis via mitochondrial and endoplasmic reticulum pathways [56]. Consistent with this, our integrated transcriptomic and metabolomic analysis revealed significant alterations in lipid metabolism, further linking oxidative stress to hepatocellular injury. Moreover, the notable up-regulation of inflammation-associated lipid mediators—particularly prostaglandin E3 in the DEHP group—indicates activation of inflammatory signaling pathways and an enhanced immune response in hepatic tissues [57]. Together, these findings underscore the critical role of oxidative stress in mediating liver damage induced by MPs and DEHP.

### 3.3. Effects of MP and DEHP Exposure on Disruption of Amino Acid Metabolism of Mice

Amino acids (AAs), as the fundamental building blocks of tissue proteins, are essential for maintaining the physiological homeostasis of organisms [58]. In particular, non-essential amino acids—such as arginine, proline, glycine, and glutamine—play critical regulatory roles in cellular signaling and various metabolic pathways. They influence nutrient absorption and disease progression by modulating gene expression, enhancing immune responses, promoting antioxidant activity, and supporting neural and skeletal muscle functions [59].

In this study, transcriptomic and metabolomic analyses revealed significant disruptions in amino acid metabolic processes following exposure to MPs, DEHP, and their combination. Altered pathways included arginine and proline metabolism, lysine degradation, lysine biosynthesis, and arginine biosynthesis. Notably, metabolites such as 5-aminovaleric acid and glutaconic acid were markedly elevated in the MP-exposed group, indicating abnormal amino acid catabolism. These disruptions suggest that MPs may interfere with nitrogen balance, energy metabolism, and key signaling networks by modulating the levels of non-essential amino acids—compounds essential for immune regulation, oxidative defense, and neurological function [60,61,62]. Importantly, glutaconic acid, an intermediate in tryptophan and lysine metabolism, has been associated with neurotoxic conditions such as glutaric aciduria type I (GA I) [62]. Its elevation in this study further underscores the potential systemic and neurological consequences of metabolic dysregulation induced by exposure to environmental pollutants like MPs and DEHP.

### 3.4. Effects of MP and DEHP Exposure on Lipid Metabolism Disorders and Membrane Integrity of Mice

This study demonstrates that exposure to MPs, DEHP, and their combined exposure significantly disrupts lipid metabolism in the liver of mice. Multiple interconnected mechanisms contribute to this disruption, including the activation of inflammation-related pathways, lipid metabolism, and altered transcriptional regulation.

In the MP-exposed group, elevated levels of hydroxylated lecithin and LysoPA (20:2) suggest the activation of inflammation-associated lipid metabolic pathways. LysoPA, a key pro-inflammatory lipid, can directly trigger immune responses [63], which is consistent with previous findings that microplastics promote lipid peroxidation [64]. Prostaglandin E3, as a prostaglandin of series 3 produced from the metabolism of eicosapentaenoic acid (EPA), plays an important role in regulating immune responses and inflammation [57]. The significant increase in prostaglandin E3 in the DEHP group highlights an up-regulation of inflammatory lipid mediators, indicating immune activation and enhanced inflammatory signaling in liver tissues.

At the same time, DEHP-induced lipid metabolic abnormalities were strongly associated with disruptions in phospholipid metabolism, particularly involving glycerophospholipids—key constituents of cellular membranes that also regulate protein recognition, signal transduction, and a wide range of cellular processes including adhesion, apoptosis, and homeostasis [65,66]. Our findings showed that MPs, DEHP, and their combination significantly altered metabolites such as 2-stearoyl-glycerol-3-phosphoglycerol, 1-(11Z-eicosenoyl)-glycerol-3-phosphate, and lysophosphatidic acids (LysoPAs), all of which are involved in glycerophospholipid metabolism. These alterations reflect the disruption of membrane structure and function. Notably, 2-stearoyl-glycerol-3-phosphoglycerol, a metabolite related to phosphatidylglycerol metabolism, was significantly down-regulated in the DEHP group, which may indicate disrupted mitochondrial membrane phospholipid synthesis, resulting in membrane structural damage and impaired energy metabolic function [67]. Moreover, the up-regulation of N1-(5-phosphoribosyl)-5,6-dimethylbenzimidazole, a precursor of vitamin B12, may represent a compensatory response to the metabolic dysfunction induced by DEHP exposure [68,69]. In parallel, the accumulation of flavin adenine dinucleotide (FAD) in the co-exposure group points to enhanced liver peroxidation and impaired electron transport chain activity [70]. Taken together, these findings suggest that DEHP and MPs disrupt mitochondrial structure and lipid metabolism, thereby exacerbating oxidative stress and energy imbalance in the liver.

MPs, DEHP, and their combined exposure may alter transcriptional regulation. Notably, gene expression changes in both the DEHP and combined exposure groups were enriched in developmental processes, which aligns with DEHP’s well-established role as an endocrine disruptor [71]. Additionally, GO analysis revealed that the regulation of molecular functions was significantly affected, with differentially expressed genes in the MP, DEHP, and combined exposure groups being significantly enriched in pathways related to metabolic processes, responses to stimuli, and biological regulation. These findings suggest that environmental pollutants may interfere with liver function by modulating the activity of transcription factors such as peroxisome proliferator-activated receptors (PPARs) [72]. Among them, PPARα is a key transcription factor that regulates lipid metabolism by controlling the expression of genes involved in fatty acid oxidation and transport [73,74]. Previous studies have shown that DEHP can significantly alter the expression of genes related to hepatic lipid metabolism through activation of the PPARα signaling pathway [75,76]. In our study, the DEHP group exhibited the highest number of differentially expressed genes (DEGs), which is consistent with its role as a typical endocrine disruptor and with findings by Hao et al., who reported that DEHP interferes with the endocrine system and affects multiple physiological processes [71]. Furthermore, in terms of energy metabolism, Feige et al. demonstrated that DEHP can impair liver energy homeostasis through a PPARα-dependent mechanism [77]

### 3.5. Effects of MP and DEHP Exposure on the PI3K/AKT Signaling Pathway and Liver Apoptosis

Our study highlights the significant role of the PI3K/AKT signaling pathway in liver apoptosis induced by MPs and DEHP exposure. The PI3K/AKT pathway is essential in regulating fundamental cellular functions such as cell survival, proliferation, and metabolism [78]. Notably, in our analysis, exposure to MPs, DEHP, and the combined MP + DEHP treatment led to a marked down-regulation of this pathway in liver tissues, with the most pronounced inhibition observed in the co-exposure group. This down-regulation aligns with the observed liver cell apoptosis and inflammatory responses. The PI3K/AKT pathway, through the activation of AKT, plays a crucial role in preventing cell death, and its inhibition is linked to the activation of apoptotic signals [79]. Furthermore, the PI3K/AKT pathway’s interaction with Bcl-2 family proteins, such as Bax and Bcl-2, influences mitochondrial dysfunction and caspase activation, which are central to apoptosis induction [80]. Several studies have demonstrated that DEHP can inhibit the PI3K/AKT/Bcl-2 signaling axis, promoting hepatocyte apoptosis and contributing to liver injury [81,82].

Consistently, our transcriptomic and molecular biology analysis revealed significant alterations in genes associated with this pathway, suggesting its involvement in the liver damage observed in our experimental model. Specifically, the results demonstrated that all exposure groups showed a marked down-regulation of PI3K, AKT, and Bcl-2 at both the protein and mRNA levels, indicating inhibition of the PI3K/AKT signaling pathway. Concurrently, significant up-regulation of TNF-α, Bax, and Caspase 3/8 was observed in the MP group, while the DEHP group exhibited additional up-regulation of IL-6 and Caspase 9, indicating a broader activation of inflammatory and apoptotic signaling. Importantly, comparisons between the DEHP and MP groups revealed significant differences in the expression of several markers, suggesting distinct toxicological mechanisms between the two pollutants. Furthermore, the MP + DEHP co-exposure group exhibited the most pronounced alterations across almost all indicators—except Caspase 8—when compared to both single-exposure groups, highlighting a potential synergistic effect. These findings suggest that MPs and DEHP can activate pro-inflammatory cytokines such as TNF-α and IL-6, which are known to influence the PI3K/AKT pathway and may form a feedback loop that exacerbates liver inflammation and apoptosis. Notably, TNF-α has been shown to promote IL-6 expression through PI3K/AKT activation, while IL-1 and IL-6 can independently activate this pathway, contributing to downstream apoptotic signaling [83,84,85]. Together, these results underscore the critical role of cytokine-mediated PI3K/AKT pathway inhibition in pollutant-induced liver injury and further highlight the enhanced toxicity of combined exposure.

### 3.6. Effects of MP and DEHP Exposure on Systemic Implications

Although our primary focus was on liver toxicity, the data also suggest broader systemic effects resulting from MP and DEHP exposure. Notably, KEGG pathway enrichment analysis of differentially expressed genes (DEGs) revealed significant enrichment in cancer-related pathways across all exposure groups, with the DEHP group and co-exposure group showing particularly prominent associations. Specifically, up-regulated genes in these groups were linked to pathways involved in hepatocellular carcinoma (HCC) and chemical carcinogenesis, such as the reactive oxygen species (ROS) pathway. The mechanism underlying this finding may be related to previous studies demonstrating that the DEHP metabolite MEHP induces DNA damage and microplastic-induced molecular alterations in protamine-like proteins, along with their DNA-binding activity [86,87]. Additionally, the enrichment of Alzheimer’s disease and amyotrophic lateral sclerosis (ALS)-related pathways in the DEHP group points to potential neurotoxic and neurodegenerative risks. These observations align with prior research suggesting a connection between plastic-derived chemicals and developmental neurobehavioral disorders, as well as Parkinson’s disease-like symptoms observed in animal models [17,88].

While these findings broaden our understanding of the toxicological impact of MPs and DEHP beyond hepatic injury, several limitations should be acknowledged. The current understanding of the ecotoxicological effects induced by MPs and DEHP remains at an early stage, particularly regarding the molecular mechanisms underlying neurotoxicity and DNA damage. Moreover, potential interactions between MPs, DEHP, and the host microbiota represent an emerging area of concern that may modulate systemic toxicity and thus merit detailed investigation. Another important limitation is the lack of assessment of MP and DEHP accumulation in the liver. Since the degree of toxicant accumulation in target organs can directly affect both the intensity and duration of toxic responses, systematically evaluating their hepatic accumulation in future studies will be critical to understanding their hepatotoxic mechanisms. Addressing these knowledge gaps will be essential for fully elucidating the systemic risks associated with microplastic and plasticizer exposure.

## 4. Materials and Methods

### 4.1. Chemicals and Animals

DEHP (>99%, CAS: 117-81-7) was procured from Shandong Zibo Langhui Chemical Co., Ltd. (Langhui Chem, Zibo, China). Uniform polystyrene microspheres (PS-MPs) with a particle size of 4–5 μm were purchased from Jiangsu Zhichuan Biotechnology Co., Ltd. (ZC Biotech, Nantong, China), DEHP was added to the mouse feed at a concentration of 200 mg/kg. MPs were mixed with deionized water to prepare a 10 mg/L suspension, which was stored at −4 °C and ultrasonicated before use. Mice were immediately transported to the laboratory from the Experimental Animal Center of Huazhong Agricultural University in Wuhan, Hubei, China. Forty-eight healthy, 3–4-week-old mice with an average weight of 12 ± 1.4 g were housed in specially designed cages for 8 weeks, during which they had access to ample food and water. To minimize stress induced by the experiment, all mice underwent a one-week acclimatization period in the housing environment before the formal experiment began. Healthy mice were then selected for further treatment.

### 4.2. Experimental Design and Exposure

The experiment included four treatment groups to evaluate the combined effects of microplastics (MPs) and di(2-ethylhexyl) phthalate (DEHP) on mice: a non-polluted control group (Con), an MP exposure group, a DEHP exposure group, and a co-exposure group receiving both MPs and DEHP. Specifically, the control group was fed a standard mouse diet and provided with clean deionized water. The MP group received the same standard diet, with drinking water supplemented with 10 mg/L MPs. The DEHP group was administered a diet containing 200 mg/kg DEHP and deionized water. The co-exposure group received a diet containing 200 mg/kg DEHP along with drinking water containing 10 mg/L MPs. Following an acclimation period, mice were randomly assigned to the four groups, each consisting of 6 replicates with 12 mice per group. Throughout the experiment, environmental conditions were strictly controlled: relative humidity was maintained at 60–80%, ambient temperature was maintained at 24 ± 2 °C, and a 12/12-hour light/dark cycle was implemented. Stainless-steel drinking water systems were used to prevent additional contamination, and all groups were fed twice daily. Feces were promptly removed to minimize the risk of secondary microplastic exposure. At the conclusion of the exposure period, mice were anesthetized, and fresh blood samples were collected into sterile centrifuge tubes. Following euthanasia, liver tissues were dissected, placed in separate sterile tubes, rapidly frozen in liquid nitrogen, and stored at −80 °C for subsequent biochemical, histopathological, transcriptomic, and metabolomic analyses. All animal procedures were reviewed and approved by the Ethics Committee of Huazhong Agricultural University (Ethical number: HZAUMO-2024-0327), and all protocols were conducted in accordance with the ARRIVE guidelines and the 3Rs principle to ensure the humane treatment of animals.

### 4.3. Measurement of Oxidative Stress Markers

Antioxidant biomarkers, including superoxide dismutase (SOD), catalase (CAT), malondialdehyde (MDA), lactate dehydrogenase (LDH), and total antioxidant capacity (T-AOC), were measured. A 10% liver tissue homogenate was prepared, and the supernatant was collected in tubes. The experimental methods followed the instructions of the SOD, CAT, MDA, LDH, and T-AOC detection kits from Nanjing Jiancheng Bioengineering Institute (JC Biotech, Nanjing, China).

### 4.4. Determination of Biochemical Parameters

Serum biochemical indicators, including aspartate aminotransferase (AST), alanine aminotransferase (ALT), urea nitrogen (UREA), creatinine (CREA), creatine kinase (CK), and alkaline phosphatase (ALP), were measured. Mouse serum supernatant was collected in tubes and analyzed using a Mindray biochemical analyzer (Mindray Biotech, Shenzhen, China).

### 4.5. H&E Staining

A portion of fresh mouse liver tissue was fixed in 4% paraformaldehyde for over 24 h. The tissue was then embedded in paraffin, cooled, and sectioned into 6 μm slices using a microtome (RM2016, Junjie Electronics Co., Ltd., Wuhan, China). The sections were stained with hematoxylin and eosin (H&E) and observed under a Nikon Eclipse C microscope (Nikon, Shinagawa, Japan).

### 4.6. RNA Sequencing and Transcriptomic Analysis

Three tissue samples from the Con, MP, DEHP, and MP + DEHP groups were randomly selected for RNA-seq analysis. Total RNA was extracted using the TRIzol^®^ Reagent isolation kit (Invitrogen, Carlsbad, CA, USA). RNA concentration and purity were measured using a Nanodrop2000 (Thermo Scientific, Waltham, MA, USA), and RNA integrity was assessed by agarose gel electrophoresis and an Agilent 2100 Bioanalyzer (Agilent Technologies, Santa Clara, CA, USA). Libraries were constructed using the Truseq^TM^ RNA Sample Prep Kit (Illumina, San Diego, CA, USA). Briefly, DNA was removed from the samples using DNase I (Thermo Scientific, Waltham, MA, USA). mRNA was isolated from total RNA using magnetic beads with Oligo(dT), fragmented, and reverse-transcribed into cDNA using random primers. The adapter-ligated products were purified and size-selected, followed by PCR amplification to obtain the final library. Sequencing was performed on the Illumina NovaSeq6000 platform (Illumina, San Diego, CA, USA).

### 4.7. UHPLC-MS/MS Metabolomic Analysis

Six tissue samples from the Con, MP, DEHP, and MP + DEHP groups were randomly selected for untargeted metabolomic analysis. Fifty milligrams of solid sample were placed in a 2 mL centrifuge tube with a 6 mm grinding bead. Four hundred microliters of extraction solvent (methanol/water = 4:1, *v*:*v*) containing 0.02 mg/mL internal standard (L-2-chlorophenylalanine) were added for metabolite extraction. The sample solution was ground in a frozen tissue grinder for 6 min (−10 °C, 50 Hz), followed by low-temperature ultrasonic extraction for 30 min (5 °C, 40 kHz). The samples were then kept at −20 °C for 30 min, centrifuged for 15 min (4 °C, 13,000× *g*), and the supernatant was transferred to injection vials with inserts for analysis. UHPLC-MS/MS analysis was performed using a UHPLC-Q Exactive HF-X system with electrospray ionization (ESI) in both positive and negative ion modes by Shanghai Majorbio Bio-pharm Technology Co., Ltd., Shangai, China (Majorbio, Shanghai, China). After analysis, the LC-MS raw data were imported into the metabolomics processing software Progenesis QI version 2.4 (Waters, Milford, MA, USA), and MS and MS/MS spectra were matched against the HMDB (https://www.hmdb.ca/, accessed on 11 May 2025) and Metlin (https://metlin.scripps.edu/, accessed on 11 May 2025) databases, as well as the Majorbio in-house database, to identify metabolites. The data matrix was uploaded to the Majorbio Cloud Platform (https://www.majorbio.com/, accessed on 11 May 2025) for further analysis.

### 4.8. Integrated Analysis of Transcriptomics and Metabolomics

For the integrated analysis of differentially expressed genes (DEGs) and differentially expressed metabolites (DEMs), three tissue samples were selected based on the dispersion in the PLS-DA model from six independent metabolomic samples for integration with transcriptomic data. First, Pearson correlation coefficients were calculated to explore the relationships between DEGs and DEMs. DEGs and DEMs were ranked from high to low based on absolute correlation coefficients. The top 20 DEGs and DEMs were selected to generate a heatmap. Subsequently, KEGG pathway analysis was applied to identify common metabolic pathways between DEGs and DEMs.

### 4.9. Western Blot Analysis

Proteins were extracted from mouse liver tissues. After total protein extraction, protein concentrations were determined using a BCA assay kit (Meilunbio, Dalian, China). Equal amounts of protein were separated on 8–12% SDS-PAGE gels and transferred to PVDF membranes. The membranes were blocked with 5% milk in TBST for 2 h at 37 °C, followed by incubation with primary antibodies at 4 °C overnight. Antibody dilution methods are listed in Appendix A. After washing three times with TBST, the membranes were incubated with horseradish peroxidase (HRP)-conjugated and secondary antibodies (1:10,000 dilution; Abclonal Biotec, Beijing, China) for 2 h. Protein bands were visualized using an ECL kit (Pumei Biotec, Wuhan, China) and scanned with a gel imaging system (Nikon, Shinagawa, Japan).

### 4.10. Quantitative Real-Time PCR (qRT-PCR) Analysis

qRT-PCR was performed using the Light Cycler 96 system (Roche, Basel, Switzerland). Reactions were carried out in 10 μL volumes, and non-specific amplification was excluded before calculating the relative mRNA expression levels. The relative mRNA expression level of the target gene was calculated using the following formula: relative mRNA expression = 2^−ΔΔCT^. The primer sequences used in this experiment are listed in Appendix A.

### 4.11. Statistical Analysis

Statistical analyses were conducted using GraphPad Prism 9.0 and one-way analysis of variance (ANOVA). Differences between groups labeled with different letters were considered statistically significant at *p* < 0.05, while groups sharing the same letter were considered not significantly different (*p* > 0.05). Additionally, the R package (version 1.6.2) was employed to perform principal component analysis (PCA) and orthogonal partial least squares–discriminant analysis (OPLS-DA) on the preprocessed data matrix. Significant differential metabolites were identified based on variable importance in projection (VIP) values from the OPLS-DA model, and *p*-values came from Student’s *t*-test, with VIP > 1 and *p* < 0.05 considered statistically significant. Differential metabolites were annotated using the Kyoto Encyclopedia of Genes and Genomes (KEGG) database (https://www.kegg.jp/kegg/pathway.html, accessed on 11 May 2025) to determine their associated biological pathways. Pathway enrichment analysis was carried out using the scipy.stats package in Python 3.9, and the most relevant biological pathways were identified using Fisher’s exact test.

## 5. Conclusions

In this study, we investigated the toxicological effects of microplastics (MPs), di(2-ethylhexyl) phthalate (DEHP), and their combined exposure on mice, focusing on oxidative stress, transcriptomics, metabolomics, and liver apoptosis. Chronic exposure to MPs and DEHP disrupted the antioxidant defense system, altered gene and metabolite profiles, and significantly affected metabolic pathways in the liver, particularly the PI3K-Akt signaling pathway, amino acid metabolism, and lipid metabolism. Notably, the combined exposure produced more pronounced toxic effects than single exposures, indicating possible additive or synergistic interactions. These disruptions ultimately triggered inflammation and hepatocyte apoptosis. Our findings provide new insights into the joint toxicity of MPs and plasticizers and underscore the potential health and ecological risks of long-term co-exposure to these environmental pollutants.

## Figures and Tables

**Figure 1 ijms-26-04943-f001:**
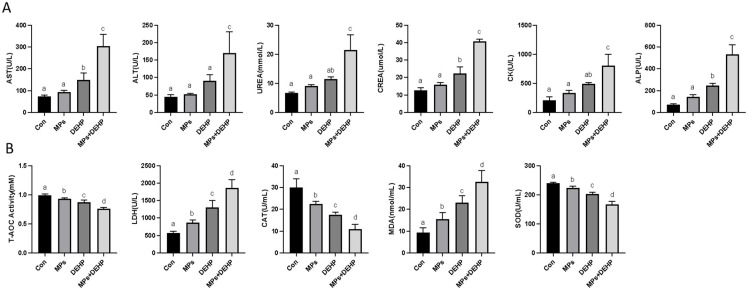
Biochemical and antioxidant measures. (**A**) Serum biochemical analysis of the effects of MP and DEHP exposure on mice. (**B**) Effects of MP and DEHP exposure on antioxidant biomarkers in mouse liver tissue. Different lowercase letters at the top of the bar indicated that the difference between the different groups was statistically significant (*p* < 0.05), and the same lowercase letter indicated that the difference was not statistically significant (*p* > 0.05). *n* = 6.

**Figure 2 ijms-26-04943-f002:**
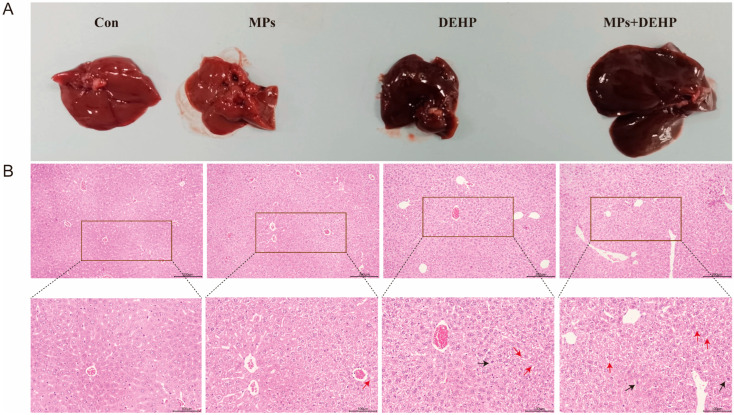
Morphological observation of the liver: (**A**) apparent of liver anatomy; (**B**) H.E. staining of the morphological changes of liver cells. The black arrow and the red arrow indicate hyperemia and vacuolization, respectively. Scale bar: 100 µm.

**Figure 3 ijms-26-04943-f003:**
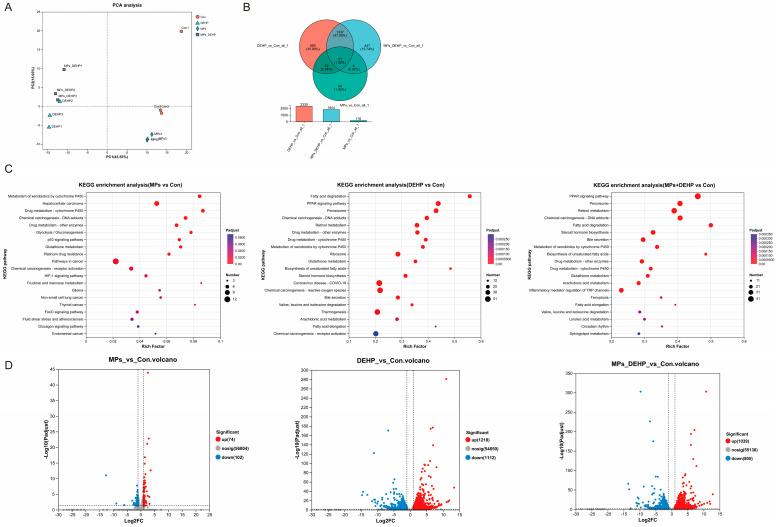
Transcriptomic analysis of mouse liver tissues after exposure to MPs, DEHP, and MPs + DEHP (*n* = 3). (**A**) PCV analysis plot of mouse gene expression levels. (**B**) Venn analysis plots of samples between MP, DEHP, and MP + DEHP treatment groups. (**C**) KEGG pathway enrichment in MPs/Con, DEHP/Con, MPs + DEHP/Con. (**D**) Volcano plot of differentially expressed genes in MPs/Con, DEHP/Con, MPs + DEHP/Con.

**Figure 4 ijms-26-04943-f004:**
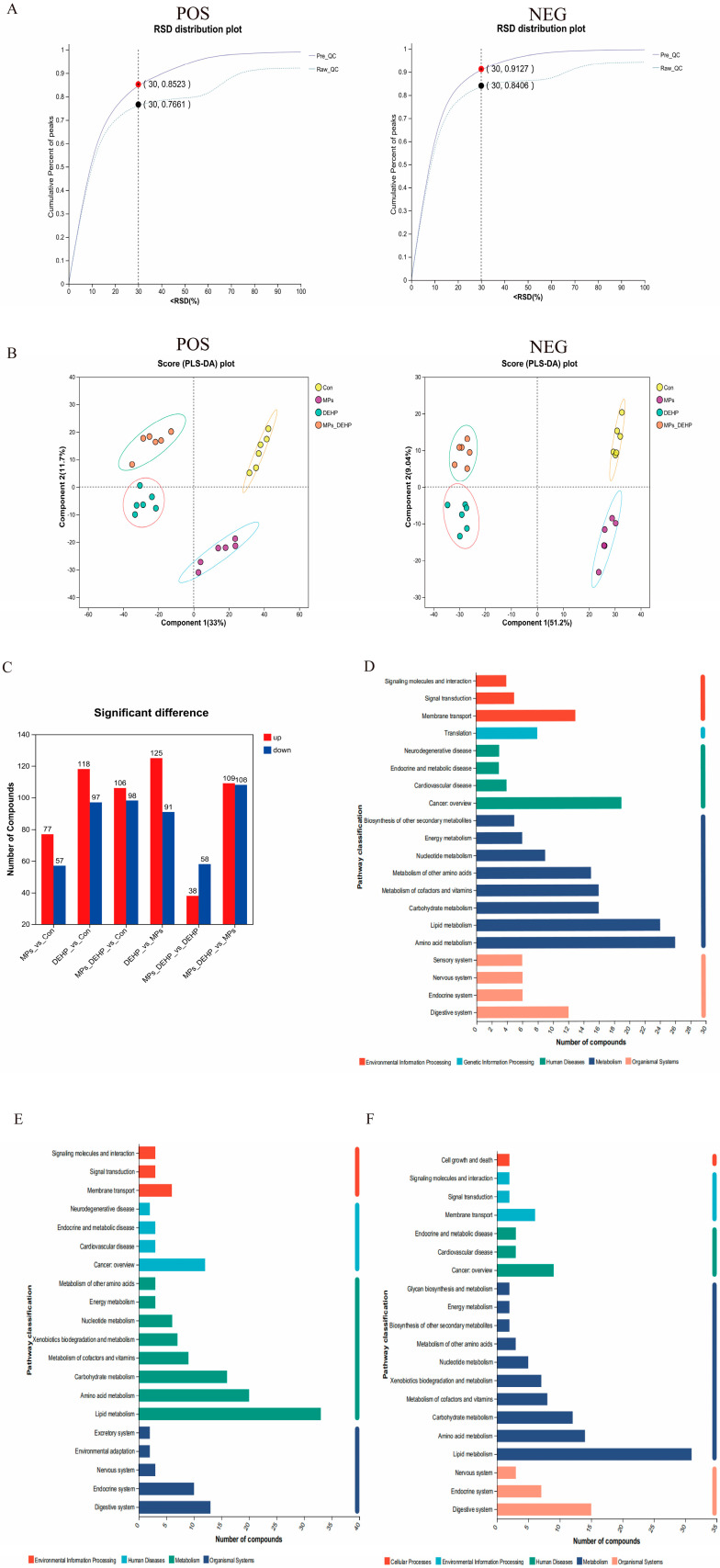
Metabolomics analysis of mouse liver tissues after exposure to MPs, DEHP, and MPs + DEHP (*n* = 6). (**A**) QC sample evaluation plot of mouse metabolomics in positive and negative ion modes, the abscordinate is the RSD (%) value, which is the standard deviation/mean, and the ordinate is the cumulative proportion of ion peaks (dashed line represents before pretreatment, solid line represents after pretreatment). (**B**) Plot of PLS-DA scores of mice after exposure to MPs, DEHP, and MPs + DEHP in positive and negative ion modes. (**C**) Number of DEMs in mice after exposure to MPs, DEHP, and MPs + DEHP. (**D**) KEGG metabolic pathways in MPs/Con group. (**E**) KEGG metabolic pathways in DEHP/Con group. (**F**) KEGG metabolic pathways in MPs + DEHP/Con group.

**Figure 5 ijms-26-04943-f005:**
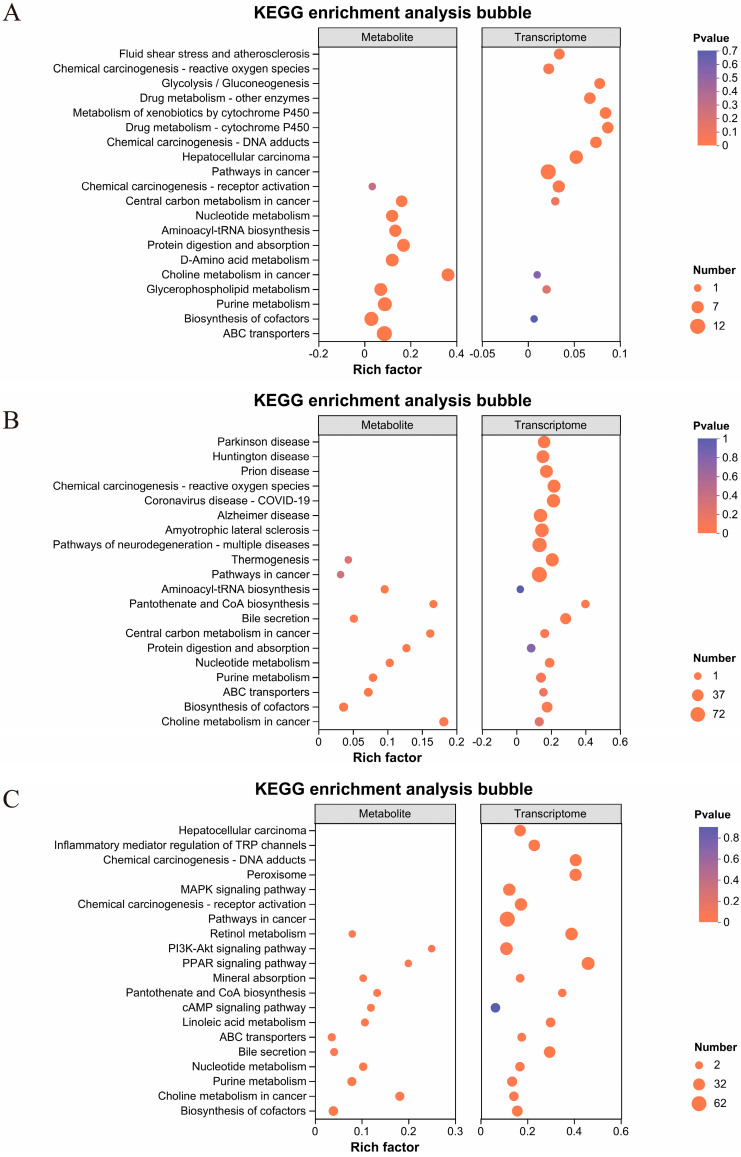
Tertiary pathways significantly enriched in genes and metabolites across the comparisons are depicted: (**A**) MPs/Con, (**B**) DEHP/Con, and (**C**) MPs + DEHP/Con. Metabolomic data are shown on the left and transcriptomic data on the right for each panel. Each bubble represents a pathway, with bubble size proportional to the number of enriched genes or metabolites.

**Figure 6 ijms-26-04943-f006:**
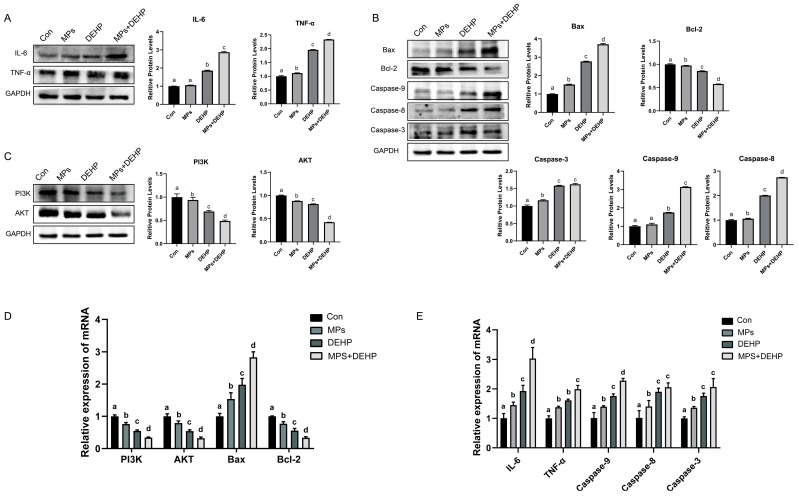
Inflammatory response and expression of PI3K/AKT pathway and apoptosis-related markers in mouse liver tissues. (**A**) Protein levels of IL-6 and TNF-α. (**B**) Protein expression of Bax, Bcl-2, and Caspase-3/8/9. (**C**) Protein levels of PI3K and AKT. (**D**) mRNA expression levels of PI3K, AKT, Bax, and Bcl-2 in mouse liver. (**E**) mRNA expression levels of IL-6, TNF-α, and Caspase-3/8/9. Groups labeled with different letters indicate statistically significant differences (*p* < 0.05), while those with the same letter are not significantly different (*p* > 0.05).

## Data Availability

The data presented in this research are included in this article; the information can be requested without problem by the people who require it.

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
