# Peer review of "Transcriptome Sequencing and Metabolite Analysis Revealed the Single and Combined Effects of Microplastics and Di-(2-ethylhexyl) Phthalate on Mouse Liver"

_ijms, 2025, doi:10.3390/ijms26104943_

Round 1

Reviewer 1 Report

Comments and Suggestions for Authors

The authors present a compelling study investigating the toxicological effects of microplastics (MPs), di-(2-ethylhexyl) phthalate (DEHP), and their co-exposure in mice, focusing on oxidative stress, apoptosis, transcriptomic, and metabolomic responses. The results indicate that MPs and DEHP induce both individual and synergistic biotoxic effects, primarily through modulation of the PI3K/AKT signaling pathway. These findings offer valuable mechanistic insights; however, several points should be addressed to enhance the clarity and rigor of the study:

  1. The manuscript would benefit from a clearer explanation of the rationale behind the selected concentrations of MPs and DEHP used in the animal experiments. Specifically, it is important to clarify whether these concentrations were optimized based on preliminary data and how they correspond to real-world environmental exposure levels.

  2. While the use of a control group is mentioned, the manuscript lacks detail on what constituted the control condition. Given that oxidative stress responses can be non-specific and triggered by various high-dose substances, further elaboration on the control setup and its justification is needed to support the validity of the observed effects.

  3. The discussion section currently presents a single, extended paragraph in each section, which can be difficult to follow. It is recommended that the authors divide this section into smaller, logically organized paragraphs to improve readability and help emphasize key findings and interpretations.

Author Response

Comments 1:[The manuscript would benefit from a clearer explanation of the rationale behind the selected concentrations of MPs and DEHP used in the animal experiments. Specifically, it is important to clarify whether these concentrations were optimized based on preliminary data and how they correspond to real-world environmental exposure levels.]

Response 1:[ We appreciate your comment regarding this issue. The rationale for selecting the concentrations of MPs and DEHP has now been included in the introduction].Page 2, Line 91-103

Comments 2:[While the use of a control group is mentioned, the manuscript lacks detail on what constituted the control condition. Given that oxidative stress responses can be non-specific and triggered by various high-dose substances, further elaboration on the control setup and its justification is needed to support the validity of the observed effects.]

Response 2: [We agree with the issue you pointed out. Additional details regarding the control group setup have been included in the Materials and Methods section] Page 3, Line 126-134

Comments 3:[The discussion section currently presents a single, extended paragraph in each section, which can be difficult to follow. It is recommended that the authors divide this section into smaller, logically organized paragraphs to improve readability and help emphasize key findings and interpretations.]

Response 3:[ "Thank you for your comment. We acknowledge that the Discussion section previously lacked coherence. We have substantially revised this section to enhance its logical flow, summarization, and overall readability] Page12-16 , Line 384-568

Reviewer 2 Report

Comments and Suggestions for Authors

In the article titled “Transcriptome sequencing and metabolite analysis revealed the single and combined effects of microplastics and di-(2- 3 ethylhexyl) phthalate on mouse liver” the authors  aimed to reveal the potential effects and mechanisms of microplastics (MPs), di-(2-ethylhexyl) phthalate (DEHP), and co-contamination of MPs and EHP(MPS+DEHP) on oxidative stress, apoptotic damage, transcriptomics, and metabolomics in mice.

I find an interesting work but some information is missing. Therefore, I suggest a major revision.

 The suggestions I would make are as follows:

  • The abstract is too long and needs to be written better, as it is a list of results with no links between one result and another. Perhaps it is better to be more general and go into more detail in the text.
  • In the abstract I would also avoid acronyms I would put in the text the first time they are mentioned.
  • The problem of microplastic is very important. Semen is so receptive that microplastics have been found. Very recently microplastics have been also found in follicular fluid: 10.1016/j.ecoenv.2025.117868. Read and quote this work in order to complete the introduction.
  • Microplastics have also been found in urine. Look for works and cite them.
  • Declare the precise modality to avoid plastic contamination during the experimental procedures
  • Specify the doses of microplastics used and whether these doses are those we can find in the environment.
  • Has the accumulation of microplastics in the liver been studied?
  • Are the changes observed in the liver necessarily related to microplastic accumulation in the liver?
  • A molecular mechanism must also be better hypothesized for the action of these microplastics
  • To support the hypothesis in the discussion of the effects of microplastics, I suggest reading and citing the following papers that describe the toxicological effects and potential risk of microplastic-induced molecular changes. This paper points to a mechanism of action of microplastics in a different system, but the papers highlight the DNA damage induced by microplastics.10.1016/j.cbi.2024.111309.
  • Some other pollutants, such as heavy metals, also induce changes in metabolomics profile. In the discussion better argue on the metabolomics alterations observed. In this regard, I suggest to read and quote works which highlight the differences in metabolites in gonad and spermatozoa of mussels exposed to some heavy metals
  • Better define the limitations of this study
  • English check is required
Comments on the Quality of English Language

The English could be improved to more clearly express the research.

Author Response

Comments 1:[The abstract is too long and needs to be written better, as it is a list of results with no links between one result and another. Perhaps it is better to be more general and go into more detail in the text.]

Response 1: [ We appreciate your feedback. The abstract has been revised to improve its overall summarization and logical flow] Page 1 , Line 8-26

Comments 2: [In the abstract I would also avoid acronyms I would put in the text the first time they are mentioned.]

Response 2: [We have recognized this issue and have revised the abbreviations used in both the abstract and the main text accordingly] Page 1 , Line 8-26

Comments 3:[The problem of microplastic is very important. Semen is so receptive that microplastics have been found. Very recently microplastics have been also found in follicular fluid: 10.1016/j.ecoenv.2025.117868. Read and quote this work in order to complete the introduction.]

Response 3:[We appreciate your insightful suggestion. Acknowledging the significance of emerging research on the reproductive effects of MPs, we have incorporated relevant literature into the Introduction to better contextualize our study]

Page 2 , Line 60

Comments 4:[Microplastics have also been found in urine. Look for works and cite them.]

Response 4:[ Thank you for your suggestion. We have revised the Introduction to include more comprehensive information regarding the effects of MPs on the urinary system] Page 2 , Line 56-57

Comments 5:[Declare the precise modality to avoid plastic contamination during the experimental procedures.]

Response 5:[ We appreciate your observation. Accordingly, we have supplemented the Materials and Methods section with additional details pertaining to this aspect] Page 3-4, Line 138-140

Comments 6:[Specify the doses of microplastics used and whether these doses are those we can find in the environment.]

Response 6:[ We appreciate your comment. The Introduction section has been revised to include the rationale behind the selected concentrations of MPs and DEHP] Page 2, Line 91-103

Comments 7:[Has the accumulation of microplastics in the liver been studied?]

Response 7:[ We appreciate your insightful comment. We recognize that this represents a limitation of our study, as our research primarily focused on hepatic injury. We intend to investigate this aspect further in our future work]

Comments 8:[Are the changes observed in the liver necessarily related to microplastic accumulation in the liver?]

Response 8:[ We appreciate your question. As noted in our response to the previous comment, this represents a limitation of our current study. We are committed to addressing this gap in our future work]

Comments 9:[A molecular mechanism must also be better hypothesized for the action of these microplastics]

Response 9:[ We appreciate your comment. This aspect has been further discussed and clarified in the Discussion section] Page 13-16, Line 430-568

Comments 10:[To support the hypothesis in the discussion of the effects of microplastics, I suggest reading and citing the following papers that describe the toxicological effects and potential risk of microplastic-induced molecular changes. This paper points to a mechanism of action of microplastics in a different system, but the papers highlight the DNA damage induced by microplastics.10.1016/j.cbi.2024.111309.]

Response 10:[ Thank you for your suggestion. We have incorporated this content into the Discussion section] Page 16, Line 547-550

Comments 11:[Some other pollutants, such as heavy metals, also induce changes in metabolomics profile. In the discussion better argue on the metabolomics alterations observed. In this regard, I suggest to read and quote works which highlight the differences in metabolites in gonad and spermatozoa of mussels exposed to some heavy metals.]

Response 11:[ We appreciate your comment. As you correctly noted, the description of metabolic or transcriptomic changes in our Discussion section was insufficiently detailed. We have now undertaken a major revision to present the omics changes in a clearer and more logical manner] Page 13-16, Line420-555

Comments 12:[Better define the limitations of this study.]

Response 12:[ We appreciate your comment. A discussion of the study's limitations has been included at the conclusion of the Discussion section] Page 16, Line556-568

Comments 13:[English check is required.]

Response 13:[ e appreciate your comment. The manuscript has been revised and underwent professional English language editing]

Reviewer 3 Report

Comments and Suggestions for Authors

Review of the manuscript entitled: Transcriptome sequencing and metabolite analysis revealed the single and combined effects of microplastics and di-(2-ethylhexyl) phthalate on mouse liver One of the major strengths of the manuscript is the use of both metabolomic and transcriptomic analyses, which increases the reliability of the results and provides a deeper understanding of the underlying mechanisms of toxicity. Moreover, the toxic effects of microplastics and phthalates represent a highly relevant topic from the perspective of environmental and public health. However, some corrections and comments need to be addressed by the authors: I believe the abstract and introduction are well-prepared. However, the introduction could be improved by explicitly stating the study's objective, such as: "The aim of the present study is to..." A clearly defined objective would enhance the reader’s understanding of the authors' intentions. I could not find any table listing the primer sequences or their catalog numbers. Similarly, the manuscript lacks information on the dilution ratios of all antibodies and their catalog numbers, which are necessary for reproducibility. The manuscript contains numerous typographical errors. For example, in line 16, "DEHPP" should be corrected to "DEHP". In the qPCR results section, the meaning of the reported numbers is unclear. It appears the authors refer to changes in gene expression. If so, they should specify whether gene expression changed by, for example, 176-fold or 176%. In scientific writing, numbers must always be reported with appropriate units; otherwise, they are meaningless. Throughout the manuscript, the authors refer to Tables 1, 2, 3, and 4. However, none of these tables are included in the manuscript or the supplementary files. This is a serious oversight. The manuscript, in its current form, lacks critical components and appears incomplete. There is no need to divide the discussion into numerous sub-sections. A more concise and integrated discussion would be more effective. The authors state that the selected doses represent “realistic environmental concentrations,” but no supporting references are provided for the doses of MPs and DEHP used. This claim requires proper citation. The discussion section is overly long and includes excessive literature review, which dilutes the main message of the study. The authors should clearly separate their own findings from those of others to improve clarity and impact. The manuscript would benefit from clarification and more precise reporting of methods and results, especially in relation to PI3K/AKT pathway analysis, metabolomics, and cytokine measurements. Additionally, the discussion should be better organized and distinguish between the authors' own data and literature references.

Author Response

Comments 1:[I believe the abstract and introduction are well-prepared. However, the introduction could be improved by explicitly stating the study's objective, such as: "The aim of the present study is to..." A clearly defined objective would enhance the reader’s understanding of the authors' intentions.]

Response 1:[ We appreciate your comment. The research objectives have been included at the conclusion of the Introduction section] Page 3, Line 103-110

Comments 2:[I could not find any table listing the primer sequences or their catalog numbers. Similarly, the manuscript lacks information on the dilution ratios of all antibodies and their catalog numbers, which are necessary for reproducibility. The manuscript contains numerous typographical errors. For example, in line 16, "DEHPP" should be corrected to "DEHP".]

Response 2:[ We appreciate your comment. We have recognized this issue and have provided the relevant information in the supplementary table, along with corrections to the spelling errors]

Comments 3:[In the qPCR results section, the meaning of the reported numbers is unclear. It appears the authors refer to changes in gene expression. If so, they should specify whether gene expression changed by, for example, 176-fold or 176%. In scientific writing, numbers must always be reported with appropriate units.]

Response 3:[Thank you for your valuable comment. We would like to clarify that we did not provide specific numerical data for the qPCR results, as our primary focus was to determine whether there were significant differences between the groups. In addition, we have now provided a more detailed description of these results in the revised manuscript]

Comments 4:[otherwise, they are meaningless. Throughout the manuscript, the authors refer to Tables 1, 2, 3, and 4. However, none of these tables are included in the manuscript or the supplementary files. This is a serious oversight. The manuscript, in its current form, lacks critical components and appears incomplete. ]

Response 4:[We appreciate your comment. The table has now been included in the supplementary materials]

Comments 5:[There is no need to divide the discussion into numerous sub-sections. A more concise and integrated discussion would be more effective.]

Response 5:[ We appreciate your comment. The Discussion section has undergone a major revision to improve the clarity and logical flow of the interpretation of our results]

Page 13-16, Line 394-555

Comments 6:[The authors state that the selected doses represent “realistic environmental concentrations,” but no supporting references are provided for the doses of MPs and DEHP used. This claim requires proper citation.]

Response 6:[ appreciate your comment. We have addressed this concern by including the rationale behind the selection of MPs and DEHP concentrations in the Introduction section] Page 2, Line 91-103

Comments 7:[The discussion section is overly long and includes excessive literature review, which dilutes the main message of the study. The authors should clearly separate their own findings from those of others to improve clarity and impact.]

Response 7:[ We appreciate your comment. As indicated in our response to the issue raised about the Discussion section, we have implemented significant corrections] Page 13-16, Line 394-555

Comments 8:[The manuscript would benefit from clarification and more precise reporting of methods and results, especially in relation to PI3K/AKT pathway analysis, metabolomics, and cytokine measurements.]

Response 8:[ We appreciate your comment. As indicated in our response to the issue raised about the Discussion section, we have implemented significant corrections] Page 13-16, Line 394-555

Comments 9:[Additionally, the discussion should be better organized and distinguish between the authors' own data and literature references.]

Response 9:[We appreciate your comment. The Discussion section has been restructured to enhance readability and to more clearly distinguish between the literature and our own data] Page 13-16, Line 394-555

Round 2

Reviewer 2 Report

Comments and Suggestions for Authors

The authors addressed all my questions. I accept the manuscript in the present form

Reviewer 3 Report

Comments and Suggestions for Authors

The Authors implemented all my suggestions.